ecology

carbon, *Dolichovespula*, insect, nitrogen, trophic position, *Vespula*

**Author for correspondence:**
Jyrki Torniainen
e-mail: jyrki.t.torniainen@jyu.fi

# Different trophic positions among social vespid species revealed by stable isotopes

Jyrki Torniainen[1,2] and Atte Komonen[2]

[1]Open Science Centre, and [2]Department of Biological and Environmental Science, University of Jyvaskyla, PO Box 35, 40014 Jyvaskyla, Finland

JT, 0000-0003-2972-5438

The social vespid wasps are common insect predators and several species behave in unison in the same biotopes. It is commonly accepted that social wasps are mainly opportunistic generalist predators without differences in prey selection and hence they compete for the same food resources. Trophic positions of six vespid wasp species and their potential prey from four sites in Finland and one in the UK were evaluated using carbon and nitrogen stable isotopes ($\delta^{13}C$ and $\delta^{15}N$). The difference in isotope values indicated different trophic positions among species. In general, *Dolichovespula* spp. showed higher $\delta^{15}N$ values than *Vespula* spp., which suggests that *Dolichovespula* forage on higher trophic levels. *Dolichovespula media* (Retzius, 1783) showed the highest $\delta^{15}N$ values, whereas *Vespula vulgaris* showed the lowest. *Dolichovespula media* partly expresses apex predator-like $\delta^{15}N$ values, whereas *Vespula* species tend to forage on primary consumers. The largest species *Vespa crabro* (Linnaeus, 1758) showed also similar $\delta^{15}N$ values as *Vespula* spp. However, $\delta^{13}C$ and $\delta^{15}N$ values of *V. vulgaris* workers varied slightly during the season. This study offers novel insights about the trophic segregation in the social wasp community, suggesting specialization in diet resource utilization, especially between *Dolichovespula* and *Vespula*.

## 1. Introduction

Social wasps (Vespinae) are ubiquitous insects with typical annual life cycle where single female (queen) establishes the nest in the spring. Queen takes care of the nest by herself till the first workers emerge. After workers take over the full care of the nest, the queen focuses only on egg laying. In the late summer, reproductives (drones and new queens) emerge, mate and the old nest establisher queen dies. In consequence, the nest starts to fade out; only the new queens overwinter [1].

Several eusocial wasp species coexist in unison in the same biotopes. Workers take care of the nest by collecting wood pulp

for enlarging the envelope and fluids and protein for the larvae [1,2]. The foraging distance of worker wasps is short, on average 200 m [3]. Therefore, nearby nests—both intra- and interspecific—are likely to compete for the same resources.

In general, wasps have been assumed to behave opportunistically when preying on invertebrates. The main prey items include several invertebrate taxa including butterfly larvae, spiders, flies, bugs, orthopterans and hymenopterans (e.g. [4,5]). The only, commonly and widely accepted, factor for prey selection is the composition and abundance of local prey items. However, there are observations that wasps are not entirely interfaced with generalist consumer behaviour. Some hornet (genus *Vespa*) species, for example, are highly specialized in preying solely on vespids [6]. Traditional prey selection studies are based on direct observation of predation events or prey items carried by returning workers. Because such studies are typically short-term single-species studies in one or few locations (e.g. [4,7]), they cannot fully reveal systematic interspecific differences in diet.

Stable isotope analysis (SIA) is widely employed in ecological studies (ubiquitously [8], migratory animals [9], wasps [10–13]). Especially useful in trophic level studies in animal communities are carbon and nitrogen stable isotopes. In general, nitrogen isotope value ($\delta^{15}$N) increases roughly by 3‰ units in each trophic step from bottom to top, whereas carbon values ($\delta^{13}$C) by 1‰ units [8]. Therefore, when evaluating the proportions of consumer prey species in the diet of target species (i.e. absolute trophic positions), trophic level baseline and source values are needed. However, direct $\delta^{15}$N and $\delta^{13}$C measurements without baseline values offer strong evidence of the relative trophic level position, when sampling is conducted in the same habitat of the study species [8,14].

Wasps are an interesting group for stable isotope studies. There are several species coexisting in the same environment using apparently same resources [1,11], which provide a study platform for species diet specialization research. In fact, Chikairashi *et al*. [10] revealed diet segregation of wasps and hornets using compound specific SIA, and Brian & Brian [11] showed that tropical wasps have a complex and diverse diet segregation from generalists to more specialized species using traditional isotope inference. In addition, eusocial wasps have different castes (workers, drones and queens) and life stages (egg, larvae and adults) [1], which can have segregated nutrition strategies [12].

We hypothesized that the vespid wasps behave as general opportunistic predators, i.e. they share trophic position, in the same biogeographic region and community. Thus, we predicted that there should not be any differences in carbon and nitrogen stable isotopes among the six study species.

# 2. Material and methods

## 2.1. Sampling

Six wasp species [*Vespula vulgaris* (Linnaeus, 1758), *V. germanica* (Fabricius, 1793), *Dolichovespula media D. saxonica* (Fabricius, 1793), *D. norwegica* (Fabricius, 1781) and *Vespa crabro*] were caught with bait traps from five sites during summer and autumn in 2019 in southern Finland (sites: Husö, Hyytiälä, Nokia and Turku) and in the UK (York) (figure 1).

The sampling sites belonged to the southern boreal (Jyväskylä, Hyytiälä, Nokia), hemiboreal (Turku, Husö) or temperate zone (York). The sites were variable in terms of human influence: Finnish sites were dominated by forests, waters and suburban areas (within 1 $km^2$ surrounding the traps), whereas the York site was in more urban area without closed-canopy forests or extensive waters (see electronic supplementary material, figure S1 for aerial photos).

In York, the traps were in a private garden, and in Nokia and Turku in broadleaved riparian forests; all analysed wasps were from the same trap. In Hyytiälä and Husö, we used individuals from three traps to get an adequate sample of all species. In both sites, one trap was in a courtyard, one in a conifer forest and one in a broadleaved forest; the distance between traps was less than 300 m. In principle, this could increase variation in prey use, and thus cannot explain systematic differences among species in different locations.

The Jyväskylä sites, in which we monitored two *V. vulgaris* nests in 2020, were in sparsely populated suburban areas (electronic supplementary material, figure S2). Nest 1 was underneath a *Salix* bush in a riparian area, and nest 2 was underground on open ruderal area; both nests were surrounded by broadleaved trees.

The average daily temperature for July to September 2019 and 2020 was similar (13–16°C) in all sites. The total rainfall was generally smaller in 2019 than in 2020 (for the Finnish sites the mean per site was

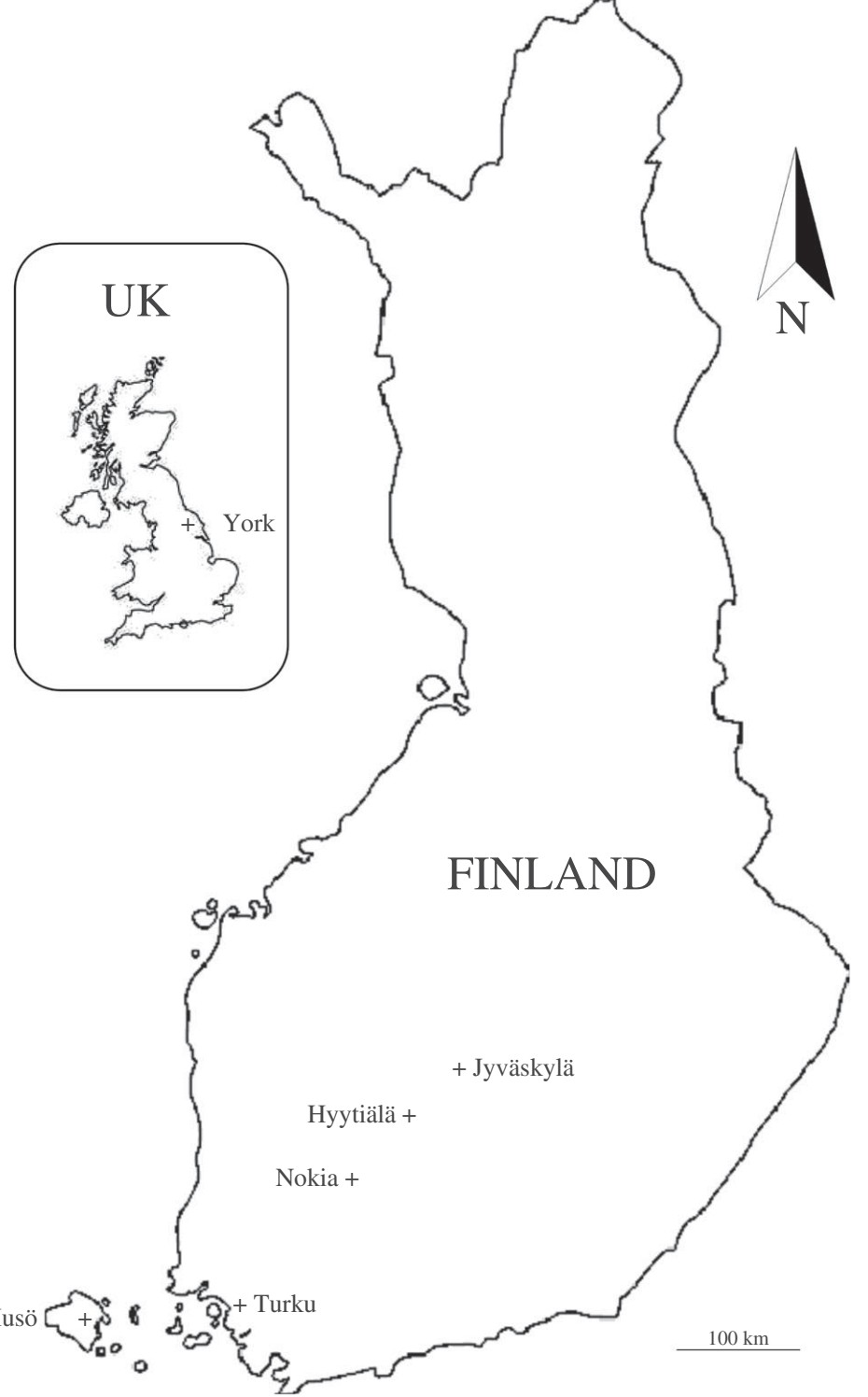

**Figure 1.** Sampling sites.

196 and 151 mm, respectively, and for the UK site 313 and 249 mm; electronic supplementary material, table S1).

In each site, three plastic containers were set approximately 1.5 m from ground on tree branches. Containers were filled with 2 dl liquid bait consisting of beer, yeast and brown sugar [15]. After 7 days, traps were emptied and refilled. Caught wasps were handled following a common practice [9], stored in ethanol and dried and identified later. Sampling was conducted on 7 July to 30 September 2019. Only worker wasps were included in the analyses to ensure that all wasps were exactly from the same habitat and used the same available resources.

In autumn 2020 (26 July to 11 September), Finnish (Jyväskylä) wasps from two separate (distance between nests approx. 2.3 km) *V. vulgaris* nests were sampled with hand net (five individuals caught in each sampling occasion; $n = 6$ and 11 sampling occasions) to evaluate the possible effect of time (i.e. late development of the nests) on $\delta^{13}C$ and $\delta^{15}N$ values (figure 4). Altogether 235 wasp individuals were analysed for $\delta^{13}C$ and $\delta^{15}N$.

In addition, in autumn 2020 potential prey animals of wasps were collected from the same sites (except York 1.8 km distance) as wasps. In total, 2–20 m transects (height 0–3 m) in the vegetation were sampled using heavy moving of hand net in the weeds and bushes. Sampled invertebrates were stored in plastic containers, frozen, dried and identified. The numbers of identified and SI-analysed animals were one to nine individuals per taxa/group in each sampling location (figure 3).

## 2.2. Isotope analysis

Wasps were dried in the room atmosphere (several weeks) or overnight in the oven (60°C). One or two hind thighs were dissected from each wasp and cut to 0.25–0.9 mg pieces into tin capsules. SIA was performed at the University of Jyväskylä, using a FlashEA1112 Elementar Analyzer connected to a Thermo Finnigan DELTA$^{plus}$ Advantage mass spectrometer (Thermo Electron Corporation, Waltham, MA, USA). Pike (*Esox Lucius*, Linnaeus, 1758) white muscle tissue was used as an internal working standard. Results are expressed using the standard δ notation as parts per thousand (‰) differences from the international standard. The reference materials used were IAEA standards of known relation to the international standards of Vienna Pee Dee Belemnite (for carbon) and atmospheric $N_2$ (for nitrogen).

In practice, potential prey animals were treated similar to wasps, although many individuals were tin capsuled as a whole due to their appropriate weight. However, in some occasions the prey animal was too large to be used as a whole, but still considered as suitable size for wasp prey. These individuals were homogenized using pestle and mortar and suitable amount of the homogenate was weighed (0.25–0.9 mg).

Precision for each run was better than 0.20‰ for $\delta^{13}C$ and 0.30‰ for $\delta^{15}N$ based on the standard deviation of replicates of the internal working standards.

## 2.3. Statistics

General linear model (GLM) was used to test whether capturing site, nest, date and/or species had effects on $\delta^{13}C$ and $\delta^{15}N$ values. Levene's test was used to ensure homogeneity of variances. One-way ANOVA or *t*-test was used to compare $\delta^{13}C$ and $\delta^{15}N$ values of different wasp species in each sampling site. *Post hoc* tests (LSD) were Bonferroni-corrected. Statistics were performed using PASW 18 and SPSS 21.0 (SPSS Inc., Chicago, IL, USA).

# 3. Results

## 3.1. Wasp thigh isotope values

There was a significant interaction between site and species on samples' $\delta^{13}C$ and $\delta^{15}N$ mean values ($\delta^{13}C$: $F = 4.85$, d.f. = 7, $p < 0.00$; $\delta^{15}N$: $F = 6.87$, d.f. = 7, $p < 0.001$).

Species mean $\delta^{13}C$ values differed only in York where *V. crabro* showed the highest value (mean = −24.57‰, s.d. = 0.35, $p < 0.001$) and *V. germanica* the lowest (mean = −25.97‰, s.d. = 0.3, $p < 0.001$) compared with the other species. Also *V. vulgaris* differed from the former species (mean = −24.15‰, s.d. = 0.75, $p < 0.001$, $p = 0.007$, respectively) (figure 2a).

Species mean $\delta^{15}N$ values differed in all sites ($F = 13.51$–221.32, $t = −14.88$, d.f. = 2–34). *Dolichovespula media* showed the highest $\delta^{15}N$ mean values in all sites and *V. vulgaris* the lowest (average difference in $\delta^{15}N$ values = 3.12‰ units, s.d. = 0.37). Also other *Dolichovespula* spp. showed higher $\delta^{15}N$ values compared with *Vespula* spp. and *V. crabro*, which showed very similar $\delta^{15}N$ values to *Vespula* spp. (figure 2b; table 1).

The potential prey items of wasps showed varying SI values. The lowest $\delta^{15}N$ values were shown in plants, but also Cicadoidea, Aphidoidea, Lepidopteran larvae and Coleoptera (herbivorous Curculionidae) showed low values (close to 0‰) when available. By contrast, ladybugs (Coleoptera: Coccinellidae) showed the highest $\delta^{15}N$ (up to 12‰) and the rest of the prey groups were between

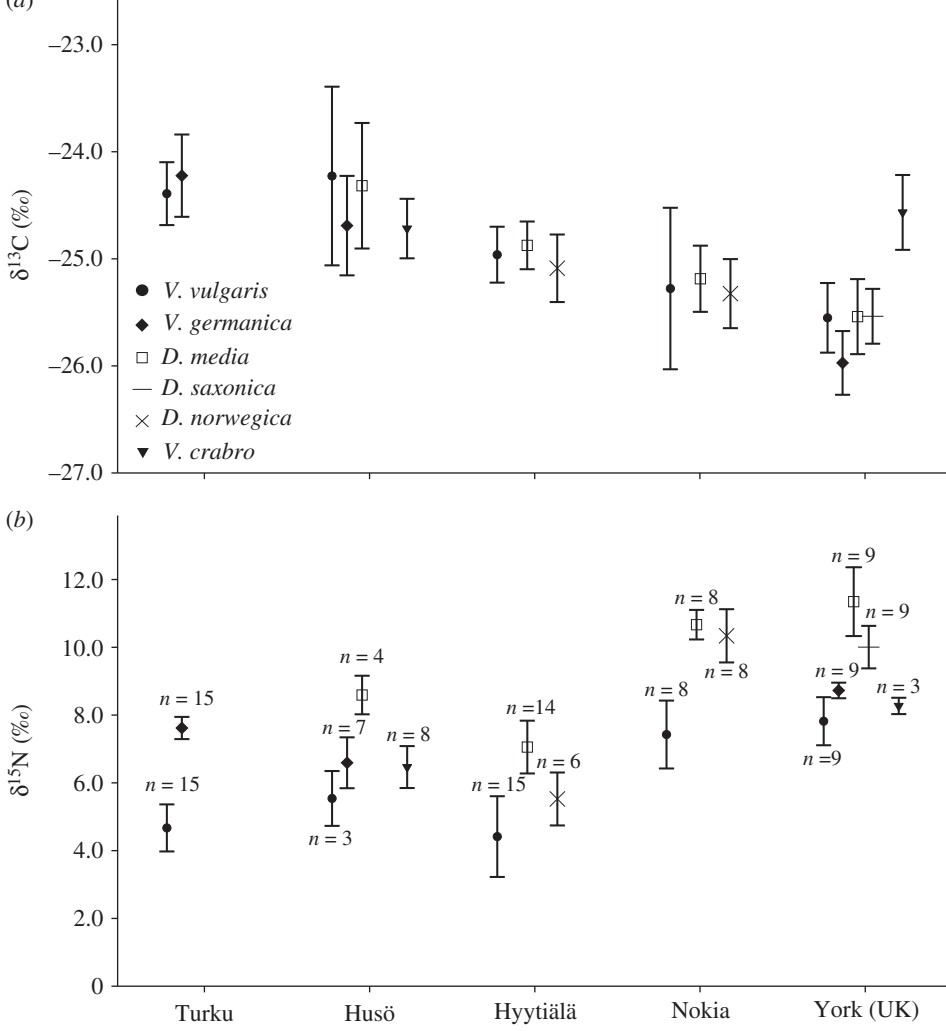

**Figure 2.** Carbon (*a*) and nitrogen (*b*) stable isotope ratios of wasps in different sampling locations (*n* = sample size). Symbols represent means and whiskers standard deviations.

those and plants. Values of $\delta^{13}C$ showed a similar pattern as in $\delta^{15}N$ values: plants, cicadas, aphids, lepidopteran larvae and coleopterans $\sim -30‰$ and rest $\sim -26‰$ (figure 3).

*Vespula vulgaris* nest workers showed significant effect of sampling date for both isotopes (GLM: $\delta^{13}C$: $F = 11.46$, d.f. = 13, $p < 0.00$; $\delta^{15}N$: $F = 7.704$, d.f. = 13, $p < 0.001$), but not for place (i.e. nest envelope: $p > 0.05$) or interaction ($p > 0.05$). SI values in one nest ranged from approximately 4 to 6‰ (N) and approximately $-27$ to $-26‰$ (C) but the other were approximately 4‰ (N) and approximately $-27‰$ (C) (figure 4).

## 4. Discussion

The significant differences in $\delta^{13}C$ and $\delta^{15}N$ values among wasp species suggest fine-scale differentiation within the predatory trophic level, i.e. utilization of different food resources. This differentiation was especially apparent in $\delta^{15}N$ values, whereas $\delta^{13}C$ values remained similar among the studied wasp species. Similarity in $\delta^{13}C$ values result mainly from the vegetation C3 carbon fixation in photosynthetic process [8], which is typical for the study sites. Differences in $\delta^{13}C$ baseline values in York are possibly related to human-induced food sources from unknown geographical origin [16,17], since the study site was in the city area. The baseline $\delta^{15}N$ value is sensitive to any organic input from external source (e.g. fertilizers) [18] and tend to raise. Wasps often aggregate in once-found food resource and use it until it is fully exploited [19,20].

**Table 1.** Differences in $\delta^{15}N$ (‰) values between wasp species in different sampling locations. Numbers are *p*-values, based on one-way ANOVA, or *t*-test in Turku. V. v, *Vespula vulgaris*; V. g., *V. germanica*; D. m., *Dolichovespula media*; D. n., *D. norwegica*; D. s., *D. saxonica*; V. c., *Vespa crabro*.

| | Turku | | Husö | | | | Hyytiälä | | | Nokia | | | York (UK) | | | | |
| --- | --- | --- | --- | --- | --- | --- | --- | --- | --- | --- | --- | --- | --- | --- | --- | --- | --- |
| | V. v. | V. g. | V. v. | V. g. | D. m. | V. c. | V. v. | D. m. | D. n. | V. v. | D. m. | D. n. | V. v. | V. g. | D. m. | D. s. | V. c. |
| V. vulgaris | <0.001 | <0.001 | | 0.038 | <0.001 | 0.060 | | <0.001 | 0.003 | | <0.001 | <0.001 | | 0.008 | <0.001 | <0.001 | 0.330 |
| V. germanica | <0.001 | — | 0.038 | | <0.001 | 0.724 | — | — | — | — | — | — | 0.008 | | <0.001 | <0.001 | 0.325 |
| D. media | — | — | <0.001 | <0.001 | | <0.001 | <0.001 | — | 0.003 | <0.001 | — | 0.407 | <0.001 | <0.001 | | <0.001 | <0.001 |
| D. saxonica | — | — | — | — | — | — | — | — | — | — | — | — | <0.001 | <0.001 | <0.001 | | 0.001 |
| D. norwegica | — | — | — | — | — | — | 0.025 | 0.003 | — | <0.001 | 0.407 | — | — | — | — | — | — |
| V. crabro | — | — | 0.060 | 0.724 | <0.001 | — | — | — | — | — | — | — | 0.330 | 0.325 | <0.001 | 0.001 | — |

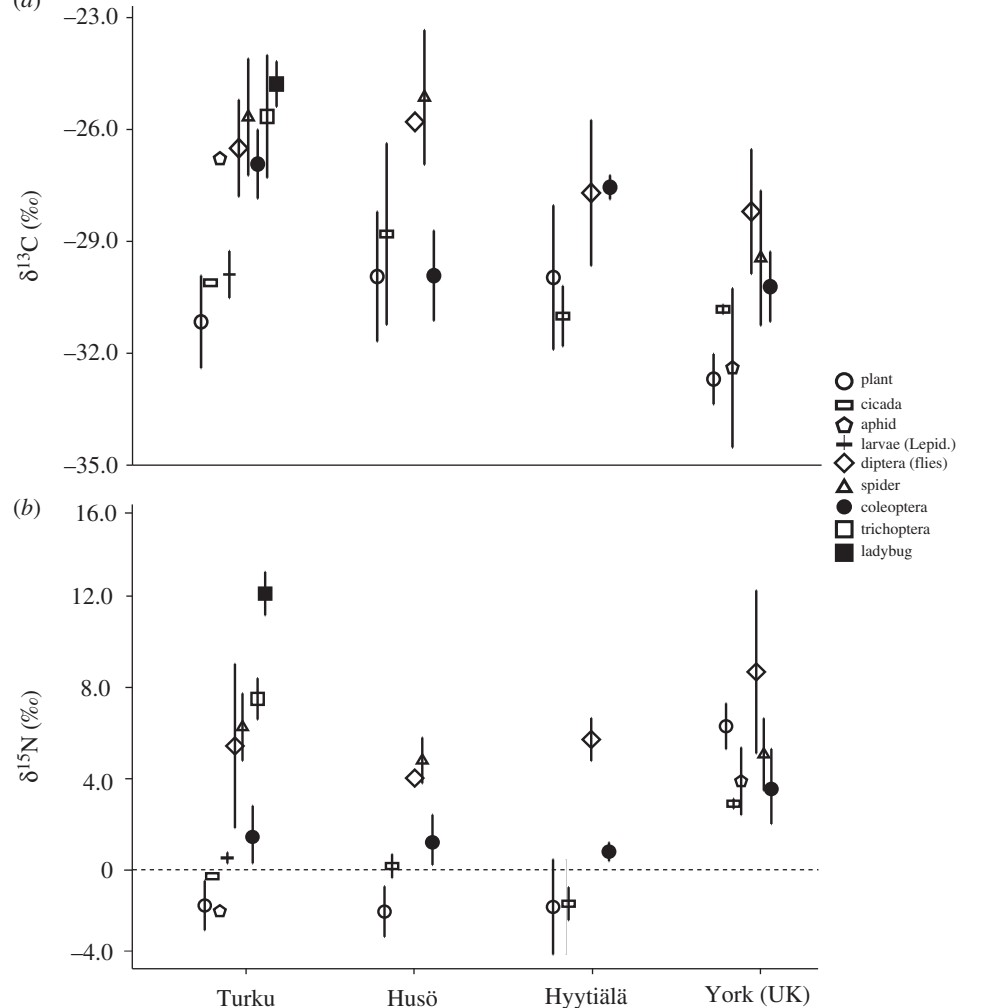

**Figure 3.** Carbon (a) and nitrogen (b) stable isotope ratios of potential wasp prey items in different sampling locations. Symbols represent means and whiskers standard deviations.

Regardless of the capturing site the most striking differences in $\delta^{15}N$ values were between *D. media* and *V. vulgaris*. *D. media* showed always the highest $\delta^{15}N$ values and, in contrast, *V. vulgaris* showed the lowest, representing clear one trophic-step difference in their trophic position. The reasoning of the difference is not straightforward, since adult wasps feed themselves mainly with different fluid sources, which usually are from the lowest trophic level (e.g. fruits and nectar) [2,21]. Therefore, $\delta^{15}N$ values of workers should be similar regardless of the species. To our knowledge, the only other possible sources for adult nutrition comes from the protein containing larval fluid exertion, which is consumed by adults (trophallaxis), or potentially from malaxation process of the protein sources carried for larvae. Presumably the flesh fluid stored in the crop of adults [22] is also passed to other workers and larvae [1]. In addition, indigestible parts are orally ejected as the so-called pellets by larvae of *D. sylvestris* [11]. Such pellets are also formed by larvae of *D. media* and *D. saxonica* [12], but not by *V. germanica* [11]. If other *Vespula* behave similarly to *V. germanica*, this might further indicate intergeneric differences in nutrition. However, the role of trophallaxis in adult wasp nutrition has been interpreted in different ways [23], but at least [21] found a clear between-species difference in amino acids of wasp larval saliva. They also stated that many of these amino acids are originating from protein sources and not from nectar. These observations together suggest between-species difference in prey selection.

Differences between both SI values were also apparent in the potential prey taxa. As expected, plants (producer baseline isotope values) showed lowest values (apart from York) with enriching continuum from various consumers to predators [8,16]. However, apart from York, $\delta^{15}N$ values showed quite clearly two groups in all other sampling sites. These groups evidently demonstrate a commonly accepted trophic cascade [8]. The first group with lowest values represent producers (plants) and their

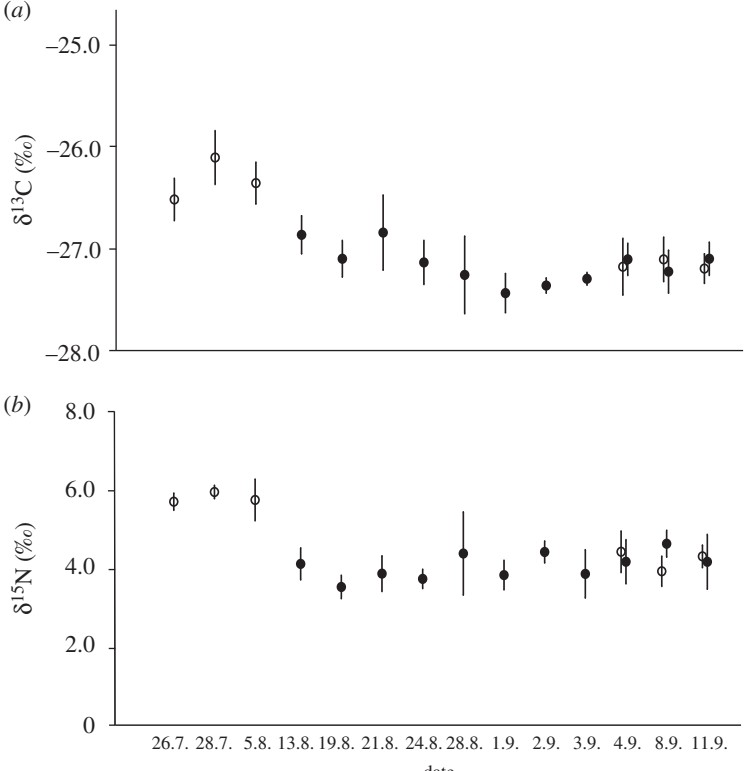

**Figure 4.** Carbon (*a*) and nitrogen (*b*) stable isotope ratios of *Vespula vulgaris* workers (*n* = 5 in each sampling date) two nests (open and closed circles). Symbols represent means and whiskers standard deviations.

consumers (aphids, butterfly larvae, cicadas, etc.), the second group with higher values comprises secondary consumers. Ladybugs were caught only in Turku. These coleopterans are assumed to prey mainly on aphids [24]. However, their very high $\delta^{15}N$ values suggest that they feed on other invertebrates as well. The $\delta^{13}C$ values of prey showed a similar two group pattern to $\delta^{15}N$ values, but difference in $\delta^{13}C$ values was too large for being one trophic step (approx. 5‰ unit difference). Isotope ecologists have commonly accepted that one trophic step in $\delta^{13}C$ values is approximately 1‰ unit [8]. Therefore, our results suggest that prey should have two sources for carbon or that our samples missed intermediate prey taxa. However, based on an overlook of the potential prey and wasp $\delta^{15}N$ values it is suggested that *Vespula* spp. and the European hornet are using prey items more from the lowest trophic level (i.e. cicadas, aphids, etc.) compared with *Dolichovespula* spp., which evidently prey more from the higher level. This is somewhat in contrast to many field studies, which have stated that both genera are generalist predators [1–3,6,25]. These field studies have been rather qualitative, i.e. they have not quantified taxon-specific prey catches for different wasp species in the same locations. By contrast, SI studies are quantitative, and the SI values are an outcome from proportions of the consumed prey [8]; thus SI approach gives a more reliable—or different at least—insight of wasp diet. For some reason, wasp $\delta^{13}C$ values are unexpectedly higher compared with prey $\delta^{13}C$ values and prevent any trophic position evaluation of different wasp species. Additional unknown source of carbon or trophallaxis-related carbon recycling process are potentially behind this striking difference in $\delta^{13}C$ values.

Especially *Vespula* spp. are thought to behave very opportunistically when foraging [1,25–27]. These wasps generally feed on carrion and human-processed protein sources, which isotopic signatures can have worldwide origin [1]. By contrast, *Dolichovespula* spp. only rarely use such sources [27,28], but could be more specialized in one or a few invertebrate taxa. For example, there are observations that *D. arenaria* have attacked predatory ladybugs (high trophic position), which are constantly avoided by *Vespula* spp., malaxated them and fed to the larvae [28]. Unfortunately, in this study ladybugs were acquired only at Turku where *Dolichovespula* were absent. However, prey source segregation is evidently true, since all *Dolichovespula* species showed higher $\delta^{15}N$ values compared with the other two genera.

Even the largest species, the European hornet *V. crabro*, which has been considered as the apex predator among wasps [1], did not show very high $\delta^{15}N$ values compared with the other wasps in the same site. In fact, the isotope values of *V. crabro* were very similar to *Vespula* spp., suggesting similar

diet. Given the larger size of the hornet, however, they could hunt larger prey [1,3] from a given trophic level, rather than switching to a higher level. In addition, *D. media* is the second largest species, which may also provide an advantage for using wider or more specific prey sources. This could be the case as high $\delta^{15}N$ values suggest feeding on higher trophic level. *Dolichovespula media* feed more on higher $\delta^{15}N$ value prey items, such as spiders (e.g. [1,2]) or other predatory species, or flies [1,2] with higher $\delta^{15}N$ values.

The striking difference in $\delta^{15}N$ values between *V. vulgaris* and *V. germanica* only at the Nokia site, where other spp. were absent, suggests a hypothesis that there could be competition between wasp species/genera. The absence of *Dolichovespula* spp. could offer a possibility for *V. germanica* to use a wider range of food sources, and vice versa, could be suppressed to compete more with *V. vulgaris* when *Dolichovespula* spp. are present. However, before any definite conclusion this hypothesis needs thorough empirical exploration.

The observed late season decline in the SI values of *V. vulgaris* nests was clear. There are two obvious explanations for this: the diet of the wasp changed or the SI values of the prey changed. It is known that wasps easily change their diet [1]. Since the nests were sampled late summer, when they start to produce queens and drones, it is possible that the change in SI values is related to the production of reproductive castes, as shown for a eusocial paper wasp [12,29]. The second explanation for the change of SI values is more unlikely, since it requires a change in baseline values. Such situation would need external input of SI matter from unknown source with clearly different values, which is unlikely, but cannot be rejected. However, changes in both SI values were not dramatic: less than commonly accepted one trophic level step [8], but still needs further investigation, including other wasp species.

SI values might be affected by the trapping protocol. The effect of rotting of wasps in traps might affect SI values and therefore mislead interpretation of trophic positions. Depleting effect of rotting on $\delta^{13}C$ and enriching on $\delta^{15}N$ (e.g. [30,31]) have been shown for tiny *Drosophila* flies, which were soaked in water for 10 days. This led to −0.81‰-unit depletion on $\delta^{13}C$ values and 0.42‰-unit enrichment on $\delta^{15}N$ values. In our study, bait traps were checked weekly, which is less than 10 days and traps were set in shadow under foliage preventing sun heat from speeding up rotting. Wasps did not express decaying appearance or smell when sampled. In addition, wasps are giants compared with tiny and delicate *Drosophila*, which slows down rotting. Furthermore, the direction of the effect should be similar in all wasp species (slightly depleted $\delta^{13}C$ and enriched $\delta^{15}N$). Based on these observations and eventually an overall small potential effect of rotting on SI values, we assume that rotting was not a significant source of error here.

Finally, it is clear that there are still many unknown factors behind the wasp nutrition, prey selection and adult–larva interaction. In this study, it was impossible to sample all suitable prey for wasps. Therefore, without recording wasp catches *in situ* it is difficult to make convincing evaluations and conclusions of wasp diet segregation. More specific dietary studies are needed to unravel the apparent diet specialization among the vespid wasp community.

Ethics. Capturing wasps was in accordance with the permits provided by the Finnish and UK animal research authorities.
Data accessibility. Dataset supporting this article was uploaded as an electronic supplementary material.
Authors' contributions. Both authors conceived the study and contributed to sampling. J.T. created the statistical analyses, figures and table, and drafted the first version of the manuscript, which A.K. further wrote and edited. A.K. created the electronic supplementary material. Both authors approved the final version of the manuscript.
Competing interests. We declare we have no competing interests.
Funding. This research was supported by award from the KONE Foundation (grant no. 21000043761).
Acknowledgements. The authors thank all persons for capturing wasps and their prey. This research was supported by award from the KONE Foundation. This study was facilitated by the nationwide sampling scheme jointly organized by the Finnish research stations (RESTAT; www.researchstations.fi). Especially thanked for sampling are Tony Cederberg and Martin Snickars (Husö Biological Station), Antti Uotila and Silja Vuorenmaa (Hyytiälä Forest Field Station), Tomi Kumpulainen, Lauri Mikonranta and Anssi Teräs.

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
