## [Peer Review File · Royal Society Open Science]

Review History

RSOS-200822.R0 (Original submission)

Review form: Reviewer 1

Is the manuscript scientifically sound in its present form?

No

Are the interpretations and conclusions justified by the results?

No

Is the language acceptable?

Yes

Do you have any ethical concerns with this paper?

No

Have you any concerns about statistical analyses in this paper?

No

Recommendation?

Major revision is needed (please make suggestions in comments)

Comments to the Author(s)

This study examined the stable N and C isotopic signatures of vespid wasp species collected in Finland and the UK. The results showed that there were significant differences in the isotopic signatures among the vespid species, suggesting trophic segregation in the wasp communities. I think that this manuscript presents new datasets on the isotopic signatures of wasps, which play an important role in many terrestrial ecosystems. Therefore, the present results would attract readers who are interested in the biology of wasps and its use as biocontrol agents for pest management. However, there are several issues to be addressed as mentioned below before recommendation could be made for publication in Royal Society Open Science.

This manuscript does not present the isotopic baseline (i.e., primary producers) and isotopic signatures of potential prey. The lack of such base isotopic values makes it difficult to appropriately interpret the isotopic data of wasps. For example, it is impossible for us to know whether the highest ^{15}N values of *Dolichovespula media* means secondary predators or primary predators without such reference datasets. Although this issue is well recognized by the authors (e.g., L53-57), I think that scientific value of "relative" trophic position should be very limited. Further, it is unclear how these wasps were sampled in the five sites. The authors would need to explain the site characteristics (e.g., vegetation), what kinds of and how many bait traps were used for each site and how many days they were put in the field. If the wasps were trapped in a liquid, they should be dried immediately after sample collection in order to prevent rotting, which is known to affect isotopic signatures. I worry about the sample drying in room temperature (L75) might have affected the isotopic signatures of the wasps. Finally, more ecological information about the wasp species should be needed to interpret the differences in their isotopic values. For example, the authors cited Sackmann et al. 2000 and Greene et al. 1976 to mention the ecological difference between the two genera *Vespula* and *Dolichovespula*, but these two articles seem to concern only one species of the two genera.

Minor comments:

Table 1 seems difficult to read. Please consider presenting the posthoc results in the Figure.

Decision letter (RSOS-200822.R0)

Dear Dr Torniaainen,

The Editors assigned to your paper RSOS-200822 "Different trophic positions amongst social vespid species revealed by stable isotopes" have made a decision based on their reading of the paper and any comments received from reviewers.

Regrettably, in view of the reports received, the manuscript has been rejected in its current form. However, a new manuscript may be submitted which takes into consideration these comments.

We invite you to respond to the comments supplied below and prepare a resubmission of your manuscript. Below the referees' and Editors' comments (where applicable) we provide additional requirements. We provide guidance below to help you prepare your revision.

Please note that resubmitting your manuscript does not guarantee eventual acceptance, and we do not generally allow multiple rounds of revision and resubmission, so we urge you to make every effort to fully address all of the comments at this stage. If deemed necessary by the Editors, your manuscript will be sent back to one or more of the original reviewers for assessment. If the original reviewers are not available, we may invite new reviewers.

Please resubmit your revised manuscript and required files (see below) no later than 07-Mar-2021. Note: the ScholarOne system will 'lock' if resubmission is attempted on or after this deadline. If you do not think you will be able to meet this deadline, please contact the editorial office immediately.

Please note article processing charges apply to papers accepted for publication in Royal Society Open Science (<https://royalsocietypublishing.org/rsos/charges>). Charges will also apply to papers transferred to the journal from other Royal Society Publishing journals, as well as papers submitted as part of our collaboration with the Royal Society of Chemistry (<https://royalsocietypublishing.org/rsos/chemistry>). Fee waivers are available but must be requested when you submit your manuscript (<https://royalsocietypublishing.org/rsos/waivers>).

Thank you for submitting your manuscript to Royal Society Open Science and we look forward to receiving your resubmission. If you have any questions at all, please do not hesitate to get in touch.

on behalf of Dr Punidan Jeyasingh (Associate Editor) and Pete Smith (Subject Editor)
openscience@royalsociety.org

Associate Editor Comments to Author (Dr Punidan Jeyasingh):

This manuscript used stable isotopes to determine the trophic positions of vespine wasp species. This represents a unique effort, and I was excited about it during pre-assessment. Due to the unique nature of the study, I had a hard time securing reviews from experts working in this area. I am grateful to the expert who assessed the manuscript. While the expert shared my enthusiasm for the motivation for the study, they picked up rather critical issues. Importantly, without baseline isotopic signatures of the diet, very little can be inferred from this data. As such, I am afraid the conclusions of the study are tenuous, and would not be of suitable for publication in RSOS. If the authors can furnish these baseline isotopic data, and reanalyze the data to account for this important source of variation in isotope signature, I would be happy to handle a revision.

Associate Editor: 2
Comments to the Author:
(There are no comments.)

Reviewer comments to Author:

Reviewer: 1

Comments to the Author(s)

This study examined the stable N and C isotopic signatures of vespid wasp species collected in Finland and the UK. The results showed that there were significant differences in the isotopic signatures among the vespid species, suggesting trophic segregation in the wasp communities. I think that this manuscript presents new datasets on the isotopic signatures of wasps, which play an important role in many terrestrial ecosystems. Therefore, the present results would attract readers who are interested in the biology of wasps and its use as biocontrol agents for pest management. However, there are several issues to be addressed as mentioned below before recommendation could be made for publication in Royal Society Open Science.

This manuscript does not present the isotopic baseline (i.e., primary producers) and isotopic signatures of potential prey. The lack of such base isotopic values makes it difficult to appropriately interpret the isotopic data of wasps. For example, it is impossible for us to know whether the highest ^{15}N values of *Dolichovespula media* means secondary predators or primary predators without such reference datasets. Although this issue is well recognized by the authors (e.g., L53-57), I think that scientific value of "relative" trophic position should be very limited. Further, it is unclear how these wasps were sampled in the five sites. The authors would need to explain the site characteristics (e.g., vegetation), what kinds of and how many bait traps were used for each site and how many days they were put in the field. If the wasps were trapped in a liquid, they should be dried immediately after sample collection in order to prevent rotting, which is known to affect isotopic signatures. I worry about the sample drying in room temperature (L75) might have affected the isotopic signatures of the wasps. Finally, more ecological information about the wasp species should be needed to interpret the differences in their isotopic values. For example, the authors cited Sackmann et al. 2000 and Greene et al. 1976 to mention the ecological difference between the two genera *Vespula* and *Dolichovespula*, but these two articles seem to concern only one species of the two genera.

Minor comments:

Table 1 seems difficult to read. Please consider presenting the posthoc results in the Figure.

===PREPARING YOUR MANUSCRIPT===

While not essential, it will speed up the preparation of your manuscript proof if accepted if you format your references/bibliography in Vancouver style (please see

<https://royalsociety.org/journals/authors/author-guidelines/#formatting>). You should include DOIs for as many of the references as possible.

===PREPARING YOUR REVISION IN SCHOLARONE===

Author's Response to Decision Letter for (RSOS-200822.R0)

See Appendix A.

RSOS-210472.R0

Review form: Reviewer 1

Is the manuscript scientifically sound in its present form?

Yes

Are the interpretations and conclusions justified by the results?

No

Is the language acceptable?

Yes

Do you have any ethical concerns with this paper?

No

Have you any concerns about statistical analyses in this paper?

Yes

Recommendation?

Major revision is needed (please make suggestions in comments)

Comments to the Author(s)

RSOS-210472

This manuscript has been much improved by adding the isotope baseline datasets and references on wasp biology. However, I think that this manuscript still needs revision. First, it would be necessary to describe more clearly what has been revealed about the trophic biology of wasps by stable isotope analysis in Introduction, and to discuss the present findings by referring to earlier

isotopic studies (e.g., Chikaraishi et al. 2011; Hyodo et al. 2011; O'Donnell et al. 2018), which have already suggested the trophic niche partitioning. Second, it would be needed to add more explanations about the materials and methods. For example, it is still not clear in what kinds of field conditions (study site size, climate conditions, vegetation, and land-use etc.) these wasps were collected. Further, the authors should discuss the potential effects of one-week beer traps on the isotope ratios. Because it is widely known that rotting could affect the isotope ratios of animals, it is common practice to use ethanol or NaCl solution to preserve invertebrates to avoid rotting. The authors should justify the use of the wasp samples soaked in the beer bait traps for the isotopic study. Finally, it should be good to perform statistical analysis on the isotope baseline data and make a convincing discussion by referring to previous isotopic studies on invertebrates. In the current form, there is almost no reference in the three paragraphs (L166-208) and therefore, the statements seem very speculative.

Minor comments:

L19: Now it should be possible to compare the ^{15}N values of wasps with those of potential prey (herbivores and predators). Please interpret the ^{15}N values of wasps and briefly explain their characteristics of trophic biology.

L22: SI values should be changed to " $\delta^{15}\text{N}$ and $\delta^{13}\text{C}$ " values.

L58: Please explain the study design here to test the authors' hypothesis (sampling at multiple sites etc.). It would be necessary to explain the reason why the wasps were collected during the summer and autumn of 2020.

L79: annual development of the nest could be investigated by sampling in autumn?

L81: Please add information about the study sites for the prey (e.g., how far from the site where the wasps were collected).

L95: Please explain what kinds of IAEA standards were used.

L102: for " $\delta^{13}\text{C}$ " and " $\delta^{15}\text{N}$ "

L106: nest and date were also included in the GLM for the wasp isotope results?

L126: Please consider conducting the statistical analyses to examine the isotopic differences across the prey samples.

L133-137: Posthoc tests could be performed to look at the differences across the sampling dates.

L145-146: Some references should be needed here.

L169: It is unclear why the authors could ascribe the incoherent ^{15}N values to human impacts. Please clarify.

L177-179: This sentence is unclear to me.

L210: What is human-made protein?

L217-218: References should be needed here.

L221: References should be needed here.

Fig.2: Stable isotope values should be modified to stable isotope "ratios".

Fig.3 and 4: Place the ^{13}C results on the upper panel to keep consistency with Fig.2. I suppose that the authors could provide more taxonomic information about the "beetle".

Chikaraishi Y, Ogawa N, Doi H, Ohkouchi N (2011) $^{15}\text{N}/^{14}\text{N}$ ratios of amino acids as a tool for studying terrestrial food webs: a case study of terrestrial insects (bees, wasps, and hornets). *Ecological Research* 26: 835-844.

Hyodo F, Takematsu Y, Matsumoto T, Inui Y, Itioka T (2011) Feeding habits of Hymenoptera and Isoptera in a tropical rain forest as revealed by nitrogen and carbon isotope ratios. *Insectes Sociaux* 58: 417-426.

O'Donnell S, Fiocca K, Campbell M, Bulova S, Zelanko P, Velinsky D (2018) Adult nutrition and reproductive physiology: a stable isotope analysis in a eusocial paper wasp (*Mischocyttarus mastigophorus*, Hymenoptera: Vespidae). *Behavioral Ecology and Sociobiology* 72.

Decision letter (RSOS-210472.R0)

Dear Dr Torniaainen

The Editors assigned to your paper RSOS-210472 "Different trophic positions amongst social vespid species revealed by stable isotopes" have now received comments from reviewers and would like you to revise the paper in accordance with the reviewer comments and any comments from the Editors. Please note this decision does not guarantee eventual acceptance.

Please submit your revised manuscript and required files (see below) no later than 21 days from today's (ie 07-Apr-2021) date. Note: the ScholarOne system will 'lock' if submission of the revision is attempted 21 or more days after the deadline. If you do not think you will be able to meet this deadline please contact the editorial office immediately.

on behalf of Dr Punidan Jeyasingh (Associate Editor) and Pete Smith (Subject Editor)
openscience@royalsociety.org

Associate Editor Comments to Author (Dr Punidan Jeyasingh):
Associate Editor
Comments to the Author:

I thank the authors for a thorough revision of the initial version. This revision was reassessed by an original reviewer. The expert is generally happy with the manuscript, but raised several constructive points that will improve the final manuscript. With much gratitude to the expert

reviewer, I invite the authors to make these adjustments and submit a fresh version. Almost there!

Reviewer comments to Author:

Reviewer: 1

Comments to the Author(s)

RSOS-210472

This manuscript has been much improved by adding the isotope baseline datasets and references on wasp biology. However, I think that this manuscript still needs revision. First, it would be necessary to describe more clearly what has been revealed about the trophic biology of wasps by stable isotope analysis in Introduction, and to discuss the present findings by referring to earlier isotopic studies (e.g., Chikaraishi et al. 2011; Hyodo et al. 2011; O'Donnell et al. 2018), which have already suggested the trophic niche partitioning. Second, it would be needed to add more explanations about the materials and methods. For example, it is still not clear in what kinds of field conditions (study site size, climate conditions, vegetation, and land-use etc.) these wasps were collected. Further, the authors should discuss the potential effects of one-week beer traps on the isotope ratios. Because it is widely known that rotting could affect the isotope ratios of animals, it is common practice to use ethanol or NaCl solution to preserve invertebrates to avoid rotting. The authors should justify the use of the wasp samples soaked in the beer bait traps for the isotopic study. Finally, it should be good to perform statistical analysis on the isotope baseline data and make a convincing discussion by referring to previous isotopic studies on invertebrates. In the current form, there is almost no reference in the three paragraphs (L166-208) and therefore, the statements seem very speculative.

Minor comments:

L19: Now it should be possible to compare the ^{15}N values of wasps with those of potential prey (herbivores and predators). Please interpret the ^{15}N values of wasps and briefly explain their characteristics of trophic biology.

L22: SI values should be changed to " $\delta^{15}\text{N}$ and $\delta^{13}\text{C}$ " values.

L58: Please explain the study design here to test the authors' hypothesis (sampling at multiple sites etc.). It would be necessary to explain the reason why the wasps were collected during the summer and autumn of 2020.

L79: annual development of the nest could be investigated by sampling in autumn?

L81: Please add information about the study sites for the prey (e.g., how far from the site where the wasps were collected).

L95: Please explain what kinds of IAEA standards were used.

L102: for " $\delta^{13}\text{C}$ " and " $\delta^{15}\text{N}$ "

L106: nest and date were also included in the GLM for the wasp isotope results?

L126: Please consider conducting the statistical analyses to examine the isotopic differences across the prey samples.

L133-137: Posthoc tests could be performed to look at the differences across the sampling dates.

L145-146: Some references should be needed here.

L169: It is unclear why the authors could ascribe the incoherent ^{15}N values to human impacts. Please clarify.

L177-179: This sentence is unclear to me.

L210: What is human-made protein?

L217-218: References should be needed here.

L221: References should be needed here.

Fig.2: Stable isotope values should be modified to stable isotope "ratios".

Fig.3 and 4: Place the ^{13}C results on the upper panel to keep consistency with Fig.2. I suppose that the authors could provide more taxonomic information about the "beetle".

Chikaraishi Y, Ogawa N, Doi H, Ohkouchi N (2011) $^{15}\text{N}/^{14}\text{N}$ ratios of amino acids as a tool for studying terrestrial food webs: a case study of terrestrial insects (bees, wasps, and hornets). *Ecological Research* 26: 835-844.

Hyodo F, Takematsu Y, Matsumoto T, Inui Y, Itioka T (2011) Feeding habits of Hymenoptera and Isoptera in a tropical rain forest as revealed by nitrogen and carbon isotope ratios. *Insectes Sociaux* 58: 417-426.

O'Donnell S, Fiocca K, Campbell M, Bulova S, Zelanko P, Velinsky D (2018) Adult nutrition and reproductive physiology: a stable isotope analysis in a eusocial paper wasp (*Mischocyttarus mastigophorus*, Hymenoptera: Vespidae). *Behavioral Ecology and Sociobiology* 72.

===PREPARING YOUR MANUSCRIPT===

===PREPARING YOUR REVISION IN SCHOLARONE===

Author's Response to Decision Letter for (RSOS-210472.R0)

See Appendix B.

Decision letter (RSOS-210472.R1)

Dear Dr Torniaainen,

It is a pleasure to accept your manuscript entitled "Different trophic positions amongst social vespid species revealed by stable isotopes" in its current form for publication in Royal Society Open Science. The comments from the Editors are included at the foot of this letter.

You can expect to receive a proof of your article in the near future. Please contact the editorial office (openscience@royalsociety.org) and the production office (openscience_proofs@royalsociety.org) to let us know if you are likely to be away from e-mail contact – if you are going to be away, please nominate a co-author (if available) to manage the proofing process, and ensure they are copied into your email to the journal.

Best regards,

on behalf of Dr Punidan Jeyasingh (Associate Editor) and Pete Smith (Subject Editor)
openscience@royalsociety.org

Associate Editor Comments to Author (Dr Punidan Jeyasingh):

I thank the authors for carefully incorporating reviewer comments. I am happy to recommend this manuscript for publication. I thank the reviewers and authors for making this peer-review process constructive!

Appendix A

Response to Editor and reviewer

Associate Editor Comments to Author (Dr Punidan Jeyasingh):

This manuscript used stable isotopes to determine the trophic positions of vespid wasp species. This represents a unique effort, and I was excited about it during pre-assessment. Due to the unique nature of the study, I had a hard time securing reviews from experts working this in area. I am grateful to the expert who assessed the manuscript. While the expert shared my enthusiasm for the motivation for the study, they picked up rather critical issues. Importantly, without baseline isotopic signatures of the diet, very little can be inferred from this data. As such, I am afraid the conclusions of the study are tenuous, and would not be of suitable for publication in RSOS. If the authors can furnish these baseline isotopic data, and reanalyze the data to account for this important source of variation in isotope signature, I would be happy to handle a revision.

Reply:

Dear Editor,

We highly appreciate that our manuscript titled: "Different trophic positions amongst social vespid species revealed by stable isotopes" (RSOS-200822) has potential to be published in *Royal Society Open Science*. Initially, the manuscript was considered as a note or short communication, since we found the results extremely intriguing and completely novel, and we thought fast publication would be the most beneficial way. However, we acknowledge that the conclusions remained relative between wasp species and explicit conclusions needed more research. Therefore, we decided to make our best at gaining samples of baseline and potential prey items from each site. In the end and to our joy, samples were provided from four sites. In addition, we added new wasp worker stable isotope data of two nests. These data show that the isotope values of wasps might change during the nest development. We hope that our additional effort makes this manuscript worth for publication in RSOS.

Reviewer comments to Author:

This study examined the stable N and C isotopic signatures of vespid wasp species collected in Finland and the UK. The results showed that there were significant differences in the isotopic signatures among the vespid species, suggesting trophic segregation in the wasp communities. I think that this manuscript presents new datasets on the isotopic signatures of wasps, which play an important role in many terrestrial ecosystems. Therefore, the present results would attract readers who are interested in the biology of wasps and its use as biocontrol agents for pest management. However, there are several issues to be addressed as mentioned below before recommendation could be made for publication in Royal Society Open Science.

This manuscript does not present the isotopic baseline (i.e., primary producers) and isotopic signatures of potential prey. The lack of such base isotopic values makes it difficult to appropriately interpret the isotopic data of wasps. For example, it is impossible for us to

know whether the highest ^{15}N values of *Dolichovespula media* means secondary predators or primary predators without such reference datasets. Although this issue is well recognized by the authors (e.g., L53-57), I think that scientific value of “relative” trophic position should be very limited. Further, it is unclear how these wasps were sampled in the five sites. The authors would need to explain the site characteristics (e.g., vegetation), what kinds of and how many bait traps were used for each site and how many days they were put in the field. If the wasps were trapped in a liquid, they should be dried immediately after sample collection in order to prevent rotting, which is known to affect isotopic signatures. I worry about the sample drying in room temperature (L75) might have affected the isotopic signatures of the wasps. Finally, more ecological information about the wasp species should be needed to interpret the differences in their isotopic values. For example, the authors cited Sackmann et al. 2000 and Greene et al. 1976 to mention the ecological difference between the two genera *Vespula* and *Dolichovespula*, but these two articles seem to concern only one species of the two genera.

Reply:

Appreciated reviewer,

Thank You for constructive advice and understanding of the significance of this work. We agree that results of stable isotope values of wasp species remained relative and final conclusions were more suggestive than firm. However, we did not suggest that *Dolichovespula* behave as a secondary predator nor that *Vespula* behaves as a primary one; in fact, we believe that trophic position (at least in vespids) is not a discrete phenomenon, but a continuous one. Actually, our aim was to offer novel knowledge of the apparent trophic continuum of the wasp species. In other terms, these wasp species select the prey targets/resources in different proportions compared to each other.

We have now tackled the missing baseline and potential prey values with more field data. Results now strengthen the evident trophic segregation between *Dolichovespula* and *Vespula*. Apparently, *Vespula* wasps utilize more the lowest trophic level compared to *Dolichovespula*. However, we acknowledge that sampled prey items could be only a fraction of the true potential prey pool in the study sites. However, trophic steps of samples are observed clearly and we believe that at least the highest and lowest levels are within the sampled items.

We have now clarified the text. We added more details of the sampling procedure (i.e. bait types, amount of baits, duration of the baiting etc.). For sure wasps were not rotten, and air drying of these tiny insects is so fast that undoubtedly wasps thighs were dry in a few hours. Therefore, it is unlikely that the drying pre-treatment is not affecting the SI results. In fact, as results show consistent difference between species SI values, the possible rotting would definitely mess that difference totally.

Ecological information of wasps is sparse. There are documented differences in prey choice among wasp species (Archer, Raveret Richter etc.), but definite conclusions about systematic intergeneric or interspecific differences would require multispecies studies in the same site and years using same methodology - we are not aware of such studies at the moment. We tried to achieve as many producer and prey items as possible, ran the SI analyses for them

and added these to the manuscript. To our knowledge, the present manuscript is the only work studying wasps' interspecific differences in prey items in the same location and year

Minor comments:

Table 1 seems difficult to read. Please consider presenting the posthoc results in the Figure.

Reply:

With all do respect, we do not understand that why the table is difficult. It is a basic matrix were to check test results and significances between wasp species in every site. Actually, reviewer's suggestion adding posthocs in addition with sampling sizes to the figure would lead more disordered view.

Sincerely,

Dr. Jyrki Torniainen

Appendix B

Response to Editor and reviewer

Associate Editor Comments to Author (Dr Punidan Jeyasingh):

I thank the authors for a thorough revision of the initial version. This revision was reassessed by an original reviewer. The expert is generally happy with the manuscript, but raised several constructive points that will improve the final manuscript. With much gratitude to the expert reviewer, I invite the authors to make these adjustments and submit a fresh version. Almost there!

Reply: Authors are thankful for this rare chance to revise manuscript once more! We have discussed and accepted majority of reviewer's highly constructive comments with serious point-to-point consideration. Some comments we found somewhat awkward, but still tried to offer a response to clarify the text. By revising the manuscript and following reviewers comments we hope the manuscript meets the high standard of *Royal Society Open Science* journal.

Reviewer comments to Author:

RSOS-210472

This manuscript has been much improved by adding the isotope baseline datasets and references on wasp biology. However, I think that this manuscript still needs revision. First, it would be necessary to describe more clearly what has been revealed about the trophic biology of wasps by stable isotope analysis in Introduction, and to discuss the present findings by referring to earlier isotopic studies (e.g., Chikaraishi et al. 2011; Hyodo et al. 2011; O'Donnell et al. 2018), which have already suggested the trophic niche partitioning. Second, it would be needed to add more explanations about the materials and methods. For example, it is still not clear in what kinds of field conditions (study site size, climate conditions, vegetation, and land-use etc.) these wasps were collected. Further, the authors should discuss the potential effects of one-week beer traps on the isotope ratios. Because it is widely known that rotting could affect the isotope ratios of animals, it is common practice to use ethanol or NaCl solution to preserve invertebrates to avoid rotting. The authors should justify the use of the wasp samples soaked in the beer bait traps for the isotopic study. Finally, it should be good to perform statistical analysis on the isotope baseline data and make a convincing discussion by referring to previous isotopic studies on invertebrates. In the current form, there is almost no reference in the three paragraphs (L166-208) and therefore, the statements seem very speculative.

Reply: We thank the reviewer of his/her constructive comments. We have now introduced the trophic biology of wasps in the Introduction with the reviewer's kind provision of the scientific articles. We knew those exist, but initially found them somewhat far from our scope (Chikaraishi was already included though), since the initial aim was a research note -type paper. However, as this manuscript has been improving all the time during the revision and received new prey data in the field of stable isotope ecology, we acknowledge that widening the perspective of wasp trophic biology via stable isotopes in the Introduction is necessary. Therefore, we have added some new introductive part.

What comes to the material and methods, we have now added thorough information of the study sites as a supplementary information. We provide maps, aerial views and more detailed description of the sites as well climate information.

Reviewer raised a concern of the potential rotting effect on isotopes values of wasps. Authors are aware of this potential and agree with the reviewer of its possibility. Therefore, we have now added a reasonable discussion of this potential effect in the Discussion section, which we, however, judge to be minor.

Discussion is now partly rewritten, with several new references.

Finally, we have acknowledged all the rest constructive comments of the reviewer below in the replies of minor comments.

Minor comments:

L19: Now it should be possible to compare the ^{15}N values of wasps with those of potential prey (herbivores and predators). Please interpret the ^{15}N values of wasps and briefly explain their characteristics of trophic biology.

Reply: Now been added some very shortly, since the character numbers in the abstract is quite limited.

L22: SI values should be changed to “ $\delta^{15}\text{N}$ and $\delta^{13}\text{C}$ ” values.

Reply: done.

L58: Please explain the study design here to test the authors’ hypothesis (sampling at multiple sites etc.). It would be necessary to explain the reason why the wasps were collected during the summer and autumn of 2020.

Reply: We are very confused about this comment. Study design should be found from the Material & Methods section in details. In addition, why should we justify the year of sampling? Basically, the year 2019 [sic!] was the year we had a chance to do this and had project funding. In the year 2020 we sampled further material because the reviewer suggested so.

L79: annual development of the nest could be investigated by sampling in autumn?

Reply: Technically yes, since ground nests are extremely difficult to find in early development. However, authors acknowledge the awkwardness of the statement: the development of the nest is in its later stage. We have now clarified this in the manuscript.

L81: Please add information about the study sites for the prey (e.g., how far from the site where the wasps were collected).

Reply: done.

L95: Please explain what kinds of IAEA standards were used.

Reply: We don’t understand what is meant here. The explanation already exists.

L102: for “delta13”C and “delta15”N

Reply: changed.

L106: nest and date were also included in the GLM for the wasp isotope results?

Reply: those variables are included and already indicated.

L126: Please consider conducting the statistical analyses to examine the isotopic differences across the prey samples.

Reply: We understand the reviewers thinking here and we carefully considered this suggestion. However, we decided not to perform a statistical analysis for the potential prey. The main reason is that there are only *potential* prey items, they are not necessarily real prey, which holds a high risk for interpreting their proportions wrong in the diet of different wasp species. The p-values do not offer anything more, but instead make the reader more confused. Secondly, the study is not about the trophic biology of an insect community, but only wasps, where SI values of baseline and potential prey provide more coherency and perspective of the isotopic space. However, reader now can see where different wasps species are “located” in the trophic cascade and evaluate the probable trophic position. To offer more details of the proportions of different wasp species diet targets, we need field observations of their true prey. When having that it would be great to perform e.g. a SIAR mixing model of the prey item SI values. This offers a great possibility for further studies in the field and in the SI lab, but at this stage authors see too high risk for too far exceeding interpretation and conclusion.

L133-137: Posthoc tests could be performed to look at the differences across the sampling dates.

Reply: Posthoc tests would reveal differences between single dates compared to each other. Therefore, authors do not understand, what is the information of the possible difference between single particular sampling days. Authors wanted to show the more important result, that SI values might change during the wasp nest development.

L145-146: Some references should be needed here.

Reply: done.

L169: It is unclear why the authors could ascribe the incoherent 15N values to human impacts. Please clarify.

Reply: clarified.

L177-179: This sentence is unclear to me.

Reply: clarified.

L210: What is human-made protein?

Reply: clarified.

L217-218: References should be needed here.

Reply: added.

L221: References should be needed here.

Reply: This requirement is confusing; why discussion of the predator size related to prey size should need a reference? However, now added.

Fig.2: Stable isotope values should be modified to stable isotope “ratios”.

Reply: changed.

Fig.3 and 4: Place the ^{13}C results on the upper panel to keep consistency with Fig.2. I suppose that the authors could provide more taxonomic information about the “beetle”.

Reply: done.

Sincerely,

Dr. Jyrki Torniaainen & Dr. Atte Komonen